# Selective dorsal rhizotomy; evidence on cost-effectiveness from England

Mark Pennington[1,2]*, Jennifer Summers[2,3], Bola Coker[2,3], Saskia Eddy[2,3], Muralikrishnan R. Kartha[1,2], Karen Edwards[4], Robert Freeman[4], John Goodden[5], Helen Powell[6], Christopher Verity[7], Janet L. Peacock[2,3]

1 Kings Health Economics, Institute of Psychiatry Psychology and Neuroscience, Kings College London, London, United Kingdom, 2 King's Technology Evaluation Centre, School of Biomedical Engineering & Imaging Sciences, King's College London, London, United Kingdom, 3 School of Population Health and Environmental Sciences, King's College London, London, United Kingdom, 4 The Robert Jones & Agnes Hunt Hospital, Oswestry, United Kingdom, 5 Department of Neurosurgery, Leeds General Infirmary, Leeds, United Kingdom, 6 National Institute for Health and Care Excellence, Manchester, United Kingdom, 7 Addenbrooke's Hospital, Cambridge, United Kingdom

* mark.w.pennington@kcl.ac.uk

**Data Availability Statement:** The individual patient data which forms the basis for our analysis was collected from children partaking in the NHS England Commissioning through Evaluation

## Abstract

### Objectives

Selective dorsal rhizotomy (SDR) has gained interest as an intervention to reduce spasticity and pain, and improve quality of life and mobility in children with cerebral palsy mainly affecting the legs (diplegia). We evaluated the cost-effectiveness of SDR in England.

### Methods

Cost-effectiveness was quantified with respect to Gross Motor Function Measure (GMFM-66) and the pain dimension of the Cerebral Palsy Quality of Life questionnaire for Children (CPQOL-Child). Data on outcomes following SDR over two years were drawn from a national evaluation in England which included 137 children, mean age 6.6 years at surgery. The incremental impact of SDR on GMFM-66 was determined through comparison with data from a historic Canadian cohort not undergoing SDR. Another single centre provided data on hospital care over ten years for 15 children undergoing SDR at a mean age of 7.0 years, and a comparable cohort managed without SDR. The incremental impact of SDR on pain was determined using a before and after comparison using data from the national evaluation. Missing data were imputed using multiple imputation. Incremental costs of SDR were determined as the difference in costs over 5 years for the patients undergoing SDR and those managed without SDR. Uncertainty was quantified using bootstrapping and reported as the cost-effectiveness acceptability curve.

### Results

In the base case, the incremental cost-effectiveness ratios (ICERs) for SDR are £1,382 and £903 with respect to a unit improvement in GMFM-66 and the pain dimension of CPQOL-Child, respectively. Inclusion of data to 10 years indicates SDR is cheaper than

programme (https://www.england.nhs.uk/commissioning/spec-services/npc-crg/comm-eval/). Ethical approval to analyse the data was provided by the National Research Ethics Service (NRES) East of England committee (REC reference 14/EE/1155). Consent for analysis of the data was sought on the basis that data would be stored securely, and access limited to the research team. As such, we are unable to upload the data to a public repository. The data is held at King's Technology Evaluation Centre (KiTEC) whose director is Steve Keevil. The data is owned by the five hospitals contributing to the database: Leeds General Infirmary, Leeds (John Goodden); Great Ormond Street Hospital for Children, London (Kristian Aquilira); Bristol Royal Hospital for Children, Bristol (Richard Edwards); Alder Hey Children's Hospital, Liverpool (Benedetta Pettorini); Nottingham University Hospitals, Nottingham (Michael Vloeberghs). The data is called the SDR database. The data on resource use for children with Cerebral Palsy is owned and resides at the Robert Jones and Agnes Hunt Orthopaedic Hospital, Oswestry. Access requests should be directed to Caroline Stewart, Manager Orthotic research & Locomotor Assessment Unit at the Robert Jones and Agnes Hunt Orthopaedic Hospital, Oswestry. The contact details for Steve Keevil are: Stephen Keevil Professor of medical physics School of Biomedical Engineering & Imaging Sciences 5th Floor, Becket House 1 Lambeth Palace Road London SE1 7EU 020 7188 3812 stephen.keevil@kcl.ac.uk The contact details for Caroline Stewart are: Caroline Stewart Senior Bioengineer/ORLAU Manager ORLAU RJAH Orthopaedic Hospital Oswestry Shropshire SY10 7AG 01691 404666 Caroline.Stewart9@nhs.net.

**Funding:** HP is employed by the National Institute for Health and Care Excellence (NICE) and was contracted by NHS England to oversee the study. MP, JS, BC, SE, MK, and JP were employed by King's College London, London, UK, in partnership with NICE and NHS England, funded by NICE, and supported by the National Institute for Health Research Biomedical Research Centre based at Guy's and St Thomas' NHS Foundation Trust and King's College London. NHS England provided funding for the intervention and the evaluation and influenced the design of the evaluation of outcomes of surgery. NHS England also paid centres to provides data to the national evaluation. The funders had no additional role in data collection or any role in the analysis, decision to publish or preparation of the manuscript. https://www.england.nhs.uk/commissioning/spec-services/npc-crg/comm-eval/.

management without SDR. Incremental costs and ICERs for SDR rose in sensitivity analysis applying an alternative regression model to cost data.

## Conclusions

Data on outcomes from a large observational study of SDR and long-term cost data on children who did and did not receive SDR indicates SDR is cost-effective.

## Introduction

Cerebral palsy (CP) is a movement disorder usually arising from disturbances in brain development prior to or around childbirth. [1] It affects 1 in 500 people in Europe. [2] A common manifestation of CP is with spasticity mainly affecting the legs (previously known as spastic diplegia). [2] Children with spasticity from CP experience muscle spasms, shortening and weakening of muscles leading to mobility challenges, pain and joint degeneration. [3] Therapies used to mitigate muscle spasms and contractions include physiotherapy, botulinum toxin (Botox), [4] intrathecal baclofen and Selective Dorsal Rhizotomy (SDR). [5]

Interest in SDR has been increasing since the 1982 publication from Professor Peacock's group. [6] Optimisation of patient selection criteria and surgical approach have led to renewed interest in SDR for treatment of lower limb spasticity in children with CP, with a particular focus on offering it to those functioning at Gross Motor Function Classification System (GMFCS) levels II and III. Children with spastic diplegic CP are assigned one of five GMFCS levels according to the impact of CP on their ability to move unaided. Children at level II are limited in their ability to run and jump or walk long distances. Children at level III typically require mobility devices to walk.

In SDR, the lumbosacral nerve rootlets are selectively severed to reduce overstimulation of muscles in the lower limbs. [7] Limited evidence from trials and observational studies links SDR with sustained improvements in physiology and anatomy. [8] However, the procedure is expensive and limited access to public funding until recently has led parents of children with CP in the UK to seek private treatment. [9]

In 2014, SDR was included in NHS England's Commissioning through Evaluation (CtE) programme. [10] The programme provides limited funding for treatments for which the evidence base is insufficient with the aim of collecting evidence on effectiveness and cost-effectiveness to inform a decision on funding. In July 2018, following an interim report on the SDR CtE scheme by King's Technology Evaluation Centre (KiTEC), NHS England agreed to commission the procedure for children aged 3 to 9 inclusive with CP with spasticity mainly affecting the legs, functioning at GMFCS levels II or III. [11] The clinical findings from the SDR CtE scheme have recently been published. [12]

As part of their evaluation of the SDR CtE scheme, KiTEC undertook a cost-effectiveness analysis of SDR. Data on outcomes following SDR were drawn from the CtE scheme. Long term cost data for children who did or didn't receive SDR was taken from data supplied by the Robert Jones and Agnes Hunt Orthopaedic Hospital (RJAH), Oswestry, UK. This unit did not participate in the CtE scheme. We estimated counterfactual data for patients undergoing SDR based on published data on children not undergoing SDR. We report the cost-effectiveness analysis here.

**Competing interests:** CV has received payments or expenses for acting as the clinical chair for NHS England Commissioning through Evaluation for SDR. MP has received personal fees from Merck. The other authors declare no competing interests. This does not alter our adherence to PLOS ONE policies on sharing data and materials.

## Methods

We undertook a cost-effectiveness analysis of SDR from a health sector perspective. The patient population was children aged 3–9 years inclusive, diagnosed with CP with spasticity mainly affecting the legs, and classified as GMFCS level II or III. The comparator consisted of usual care which included physiotherapy, orthopaedic surgery, and drug treatments to alleviate spasticity. The intervention arm includes SDR in addition to treatments provided as part of usual care. We undertook subgroup analysis according to GMFCS level.

Five centres experienced in providing SDR were recruited to the CtE scheme and followed usual clinical practice with respect to SDR surgery and post-operative physiotherapy; 60–70% of the L1 to S1 nerve rootlets were cut in the vast majority of patients. Children received 3-weeks of daily physiotherapy rehabilitation at the treating centre, prior to discharge to their community provider team. This was remunerated as part of the SDR tariff. The majority of children received at least 2–3 hours of physiotherapy a week in the first three months following surgery and at least 1–2 hours in the second three months. Enhanced community physiotherapy continued to 24 months after surgery for the majority of children. [12]

The primary outcome was the Gross Motor Function Measure (GMFM-66). GMFM-66 was designed to assess gross motor function in children with CP. [13] The instrument has 66 items across five dimensions: lying and rolling; sitting; crawling and kneeling; standing; and walking, running and jumping. Each item is scored from 0 to 3. Scores within dimensions are converted to a percentage and a summary score is calculated as the mean of percentage scores on each dimension. We also evaluated a secondary outcome, the pain and impact of disability domain (hereon 'CPQOL-pain') of the Cerebral Palsy Quality of Life Questionnaire for Children (CPQOL-Child). [14] The CPQOL-Child has seven domains and is intended to capture the impact of CP on children's wellbeing. A summary score is not available for the measure. The pain domain has 8 items; the score for each item is converted to a percentage prior to deriving the mean across items. These outcomes were selected prior to data analysis to span the main perceived benefits of SDR.

### Data sources

Data on the effectiveness of SDR was taken from the SDR CtE scheme. The scheme enrolled 137 children from 2014 to 2016 at five centres across England: Alder Hey Children's NHS Foundation Trust; University Hospitals Bristol NHS Foundation Trust; Great Ormond Street Hospital for Children NHS Foundation Trust; Leeds Teaching Hospitals NHS Trust and Nottingham University Hospitals NHS Trust. Children were eligible for SDR if they met the following criteria:

- Aged between 3 and 9 years

- GMFCS II or III

- Dynamic spasticity in lower limbs affecting function and mobility

- Absence of dystonia

- MRI showing typical CP changes and no damage to key areas of brain controlling posture and coordination–periventricular leukomalacia or white matter injury of prematurity, without involvement of thalami or basal ganglia

- No evidence of genetic or neurological progressive illness

- Mild to moderate lower limb weakness with ability to maintain antigravity postures

- No significant scoliosis or hip dislocation (Reimer's index<40%) [15]

The CtE scheme received Health Research Authority approval in September 2014 (Integrated Research Approval System (IRAS) project ID: 162253). Consent to enrol children in the study and collect outcome data was sought from parents after provision of patient information sheets to parents and children. In addition to GMFM-66 and the CPQOL-Child, data collection included gait assessment, spasticity (Modified Ashworth Scale—MAS), motor control (Boyd and Graham test) and adverse events following surgery. [16,17] Evaluation of these outcomes are reported in more detail in the clinical paper. [12] Data on GMFM-66 and CPQOL--Child were collected prior to surgery and at 6, 12 and 24-month follow-up. Data were collected by the five centres and reported to a bespoke database created using Research Electronic Data Capture (REDCap).

The SDR CtE scheme did not include the collection of data on resource use and in particular did not provide data on a comparison group of children with CP who did not undergo SDR.

We were able to access inpatient data relating to treatment for CP for 26 children at RJAH. The data consisted of all assessments and treatments for CP at RJAH including physiotherapy, surgery and Botox injections. The data did not include primary care and oral drug treatments. Hospital remuneration for each procedure was available as the 2016/17 locally agreed tariff. The data were collected over the period from 1994 to 2017 and children were included if they had a continuous record of care at RJAH for at least four years. Children referred to RJAH for consideration for SDR underwent assessments including history, examination, 3D instrumented gait analysis and discussion at an MDT meeting. Eleven patients who met the inclusion criteria set by the SDR CtE protocol did not go on to have SDR, either due to a lack of funding (4 children), or due to stricter criteria set by the RJAH team including raised BMI, or a subjective impression that the degree of weakness or selective control would result in a poorer outcome. Consent for use of their data in research had previously been obtained for each child by RJAH.

## Assessment of incremental effectiveness

The incremental effectiveness of SDR on GMFM-66 was determined as the difference in GMFM-66 observed at two year follow-up in the CtE cohort and that predicted from published growth curves. Gross motor function increases rapidly in the early years after birth as children develop. However, for children with CP the rate of increase is slower and for those with more severe disease, gross motor function may decline in adolescence. The change in GMFM-66 between baseline and two-year follow-up for children receiving SDR in the CtE scheme was assessed against expected changes in the absence of surgery for children with CP of the same age and GMFCS level using longitudinal data on GMFM-66 in a cohort of Canadian children with CP recruited over the period 1996–2001 (the CanChild cohort). These data had been analysed and developmental curves for children in each of the five GMFCS categories reported. [18] We used the published growth curve models to predict GMFM-66 at two-year follow-up in the absence of SDR, for children in the CtE scheme on the basis of age, GMFCS level and baseline GMFM-66. This approach differed slightly to the analysis of GMFM-66 in the clinical evaluation in which observed GMFM-66 scores for children at baseline and two years were compared with the distribution of scores observed in the CanChild cohort. [12] The prediction models used here allowed us to estimate GMFM-66 for each child in the absence of SDR and hence to construct a counterfactual outcome for each child undergoing SDR.

The incremental effectiveness of SDR on CPQOL-pain was estimated as the change in the domain score between baseline assessment prior to SDR and two-year follow-up.

Missing data were imputed using Multiple Imputation (MI). [19] We exploited data from intervening follow-ups at 6 and 12 months in addition to data on mobility and gait to impute missing outcome data.

Uncertainty in the estimates of effectiveness in the primary (GMFM-66) and secondary outcome (CPQOL-Child) was quantified by bootstrapping. We applied a two-stage bootstrap which allows for the clustering of data within the five centres partaking in the SDR CtE scheme. [20] Following MI, two-stage bootstrapping was applied to each imputed dataset and results were combined across the datasets using Rubin's rules. [21] Further details on the methods, including prediction of GMFM-66 and imputation of missing data, are provided in the S1 File.

## Assessment of incremental cost

The incremental cost of SDR was determined as the difference in cost for children who did and did not receive SDR in the RJAH cohort. We had pre-specified the primary analysis as the difference in costs for children who did and did not receive SDR (RJAH cohort) over ten years from assessment for SDR, with adjustment for age at assessment and GMFCS level. However, we had insufficient data beyond five years in the control group to support a robust analysis. Consequently, we present our main findings over five years and include data to ten years in a sensitivity analysis. In further sensitivity analysis, we adjusted for children declined SDR on clinical grounds. Data were considered complete for the years in which there was evidence that children were under the care of RJAH for at least 6 months. Missing costs by year were imputed using MI with Predictive Mean Matching. [22] Costs were discounted at 3.5% per annum as recommended by the National Institute of Health and Care Excellence (NICE). [23]

We investigated the distribution of the cost data with the intention of applying linear regression to adjust for case-mix provided the data were not highly skewed. The mean incremental cost of SDR and the standard error were determined by combining regression results across imputed datasets using Rubin's rules. [21] More details of the analysis of cost data are given in the S1 File.

## Assessment of cost-effectiveness

Cost-effectiveness is reported as the Incremental Cost-Effectiveness Ratio (ICER) and the Cost-Effectiveness Acceptability Curve (CEAC). The ICER is the ratio of incremental cost and incremental effectiveness and reports the cost per additional unit of health gained from the intervention. An ICER was calculated where one comparator was both more effective and more costly than the other. The CEAC plots the likelihood that an intervention is cost-effective across a range of maximum values the decision maker is prepared to pay for a unit improvement in health. It integrates the impact of uncertainty in cost and outcome data.

We simulated 1,000 estimates of the incremental cost of SDR by sampling values from a Normal distribution with mean and variance determined from regression analysis of the cost data from the RJAH cohort. We simulated 1,000 estimates of the incremental effectiveness of SDR on GMFM-66 from a Normal distribution with mean and variance determined from the bootstrap replicates of the change in GMFM-66 scores at two year follow-up in the CtE cohort; the change in GMFM-66 was calculated as the observed datum minus the predicted value from the CanChild growth curve. Each of the 1,000 incremental cost estimates was paired at random with one of the 1,000 incremental effectiveness estimates to generate 1,000 pairs. The paired data were used to generate a CEAC (details in S1 File). The same procedure was used to generate a CEAC for the secondary outcome measure, and for the subgroup analysis.

**Table 1. Baseline characteristics of patients in the CtE and RJAH cohorts.**

| | CtE cohort | RJAH cohort | |
| --- | --- | --- | --- |
| | n = 137 | SDR (n = 15) | No SDR (n = 11) |
| Mean age at assessment in years (SD) | 6.54 (1.92) | 6.58 (1.11) | 7.41 (1.13) |
| Mean age at SDR surgery in years (SD) | 6.59 (1.92) | 7.04 (1.16) | - |
| GMFCS level II* (number) | 38.0% (52) | 33.3% (5) | 18.2% (2) |
| Wheelchair/buggy use | 67.2% | 73.3% | 90.9% |

*The remaining proportion of the cohort was GMFCS level III.

## Results

Table 1 reports baseline characteristics of children in the CtE cohort, and children in the RJAH cohort according to whether they received SDR. Children receiving SDR in the RJAH cohort were similar to children in the CtE cohort. Children who did not receive SDR in the RJAH cohort were slightly older at assessment and more likely to be GMFCS III. Table 2 reports GMFM-66, CPQOL-pain, gait analysis and measures of mobility at baseline and follow-up in the CtE cohort. Function improved over time. Pain scores improved at six months after SDR and then plateaued at subsequent assessments. Less than 10% of data were missing with the exception of gait scores. Compared to predicted GMFM-66 at 24-month follow-up the mean (SD) incremental gain is 5.2 (0.53). The mean (SD) improvement in CPQOL-pain at 24 months is 7.9 (1.82). The subgroup analysis indicates larger improvements in both GMFM-66 and CPQOL-pain in patients in GMFCS level II compared with those in GMFCS level III (details in S1 File).

Cost data were available for all children in the RJAH cohort receiving SDR up to 8 years after assessment and for 11 children (73%) at ten years. Nine children (82%) not receiving SDR had complete data at five years after assessment and two (18%) had complete data at ten years. For children with complete data, mean costs at five years were £33,282 in those receiving SDR and £31,030 in those not receiving SDR. Fig 1 displays the mean cost by year for children with non-missing data according to SDR status. Interventions received by children in the SDR group and the non-SDR group over the first five years are tabulated in the appendix. Costs in the SDR group are considerably higher in the first year, driven by the cost of the surgery and post-operative rehabilitation (£22,650). (Note: some children received SDR in the second year after the initial assessment.) At year 3 and beyond, mean costs are consistently lower each year in the SDR group.

The distribution of cost data was not strongly skewed. Table 3 presents the results of linear regression analyses to estimate the incremental cost of SDR. The base case analysis suggests the initial cost of SDR and 3-weeks' initial rehabilitation (£22,650) is partially offset at five years. Costs associated with SDR rise after adjustment for clinical criteria contraindicating SDR. Analysis of costs at ten years suggests the cost of SDR is entirely offset by reduction in

**Table 2. Gross motor function and quality of life at baseline and follow-up in the CtE cohort (n = 137).**

| | Baseline | 6 months | 12 months | 24 months |
| --- | --- | --- | --- | --- |
| Mean GMFM-66 score (% missing) | 59.0 (0%) | 61.7(0%) | 63.6 (1.4%) | 66.0 (3.6%) |
| Mean CPQOL-pain (% missing) | 36.4 (2.9%) | 25.3 (5.1%) | 28.8 (5.1%) | 27.6 (8.0%) |
| Mean CPQOL function (% missing) | 70.5 (2.9%) | 77.6 (5.1%) | 78.0 (5.1%) | 78.5 (7.3%) |

nr—not recorded.

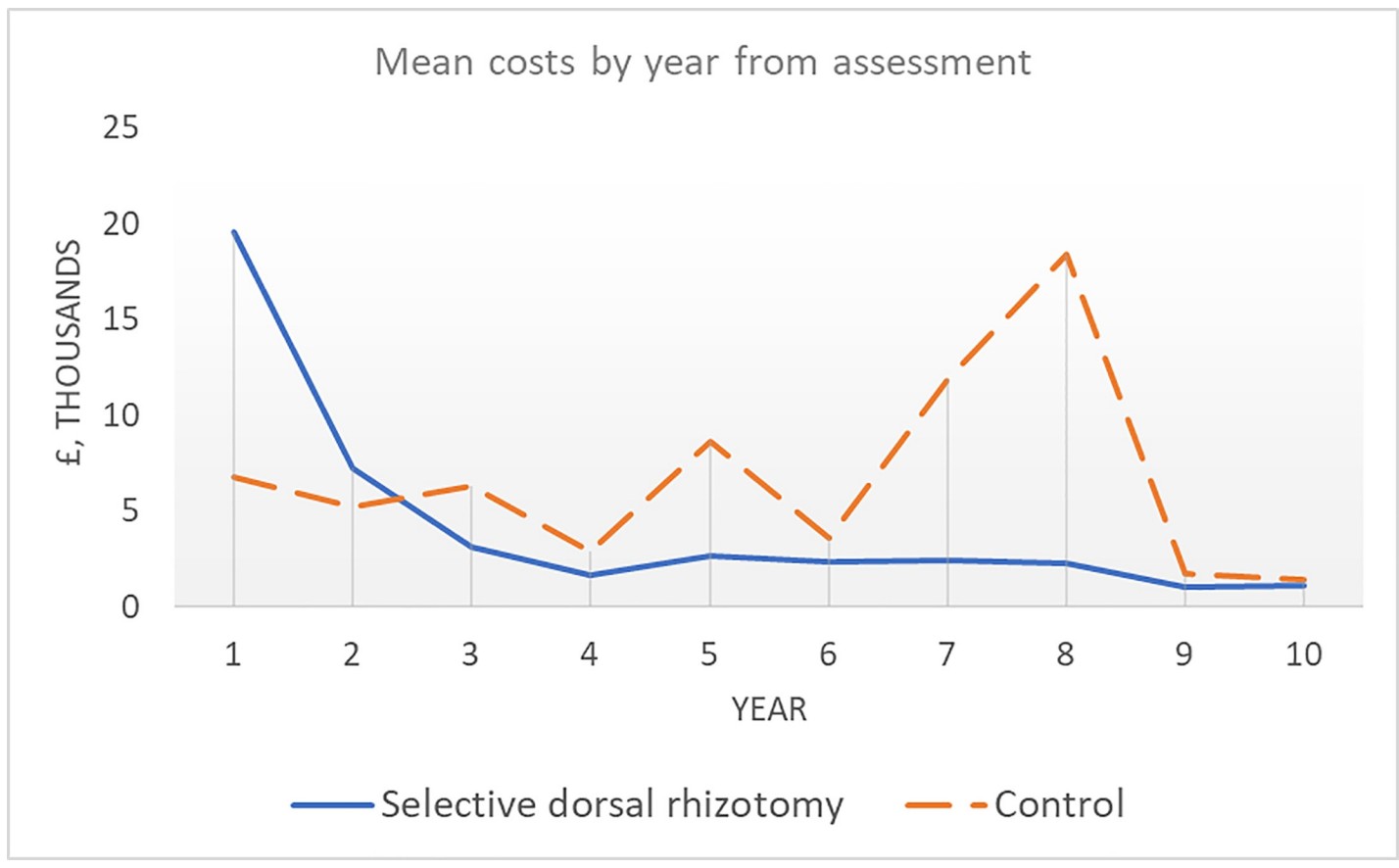

**Fig 1. Mean costs per child over time (years) by treatment status.**

the subsequent cost of supporting children. Confidence intervals are wide, and all include zero.

In the base case analysis ICERs for SDR with respect to GMFM-66 and CPQOL-pain are £1,382 and £903, respectively. This means that it costs around £1400 to generate each unit improvement in GMFM-66 and £900 to generate each unit improvement in CPQOL-pain. Figs 2 and 3 report the CEACs for the primary and secondary outcomes under the base case,

**Table 3. Estimates of the impact of treatment (SDR surgery) on mean cost per child at 5 and 10 years.**

|  | Incremental cost of SDR | 95% CI |
|---|---|---|
| Complete case, 5 years, raw | £2,252 | -£7,641 to £12,145 |
| Complete case, 5 years, adjusted* | £5,041 | -£6,057 to £16,139 |
| Imputed, 5 years, raw | £4,849 | -£5,250 to £14,949 |
| Imputed, 5 years, adjusted* | £7,160 | -£3,998 to £18,318 |
| Imputed, 5 years, sensitivity analysis# | £12,035 | -£1,982 to £26,052 |
| Imputed, 10 years, raw | -£9,132 | -£26,648 to £8,385 |
| Imputed, 10 years, adjusted* | -£5,426 | -£23,788 to £12,936 |
| Imputed, 10 years, sensitivity analysis# | £2,271 | -£24,407 to £28,950 |

*Adjusted for age and GMFCS

#Adjusted for age, GMFCS and clinical criteria contraindicating SDR.

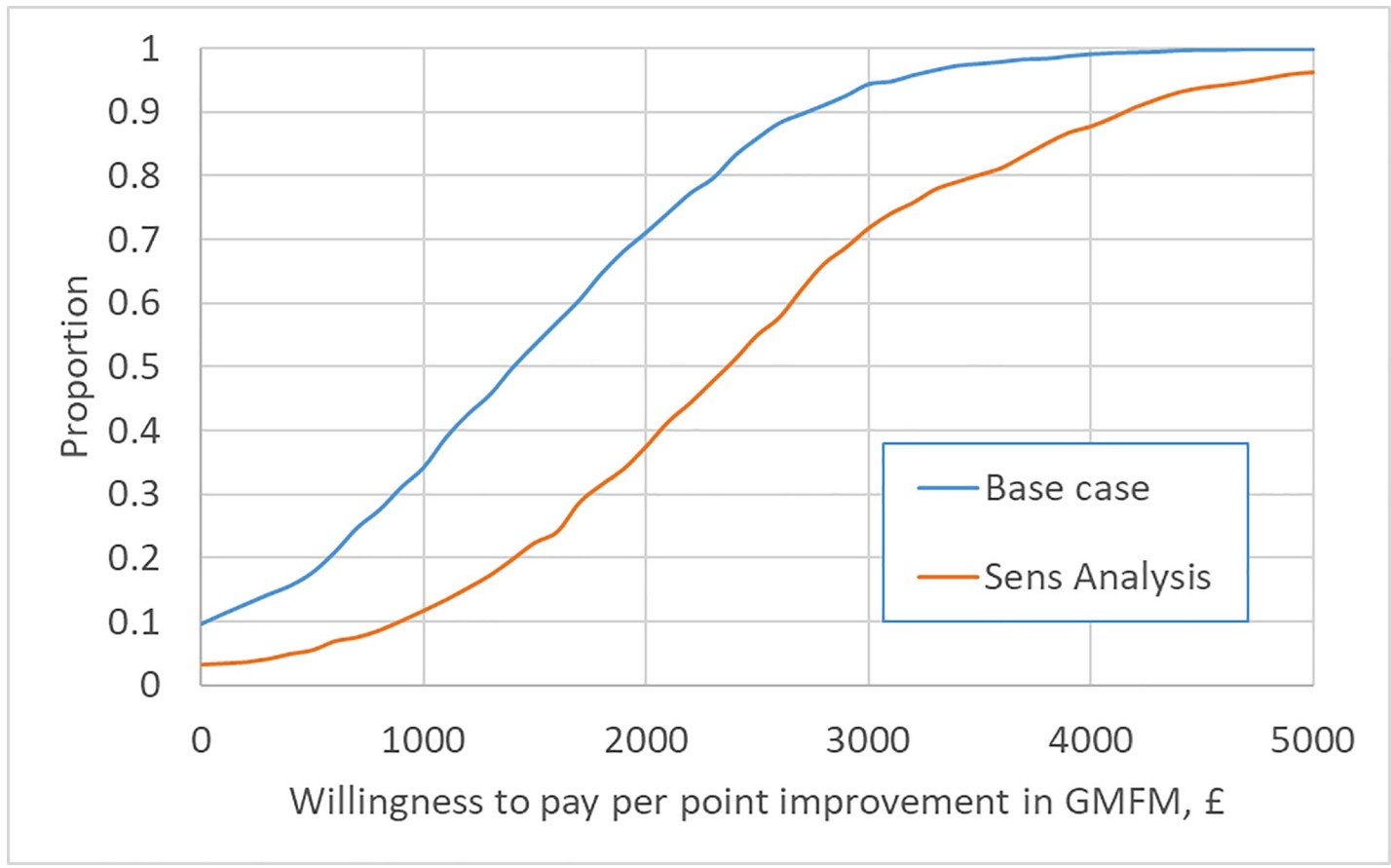

**Fig 2. Cost-Effectiveness Acceptability Curve (CEAC) for primary outcome (GMFM-66).**

and in sensitivity analysis controlling for clinical criteria contraindicating SDR. In the base case, the likelihood that SDR is cost-effective rises above 95% at a value of £3,150 for a unit gain in GMFM-66 and £2,350 for a unit reduction in CPQOL-pain. Subgroup analysis indicates a higher likelihood of cost-effectiveness for patients in GMFCS II and a lower likelihood for those in GMFCS III (CEACs are provided in the S1 File). In sensitivity analysis applying a time horizon of ten years, SDR dominates (it is cheaper and more effective than management without SDR). In sensitivity analysis controlling for clinical criteria contraindicating SDR, ICERs for SDR with respect to GMFM-66 and CPQOL-pain are £2,323 and £1,517, respectively; the likelihood that SDR is cost-effective reaches 95% at a value of £4,750 and £3,550 for unit gains in GMFM-66 and CPQOL-pain, respectively.

## Discussion

Our analysis indicates SDR in eligible children is likely to be cost-effective if decision makers value a unit gain in GMFM-66 at more than £1400 and a concomitant improvement in CPQoL-Pain at more than £900. Children in the CtE cohort showed a greater improvement in mean GMFM-66 at two-year follow-up than values predicted using data from the CanChild cohort and bootstrap resampling indicated the finding was unlikely to be due to chance. [18] Reported pain was also significantly reduced at two-year follow-up. Data from RJAH suggests that the cost of SDR is partially offset by a reduction in the costs of caring for children over a

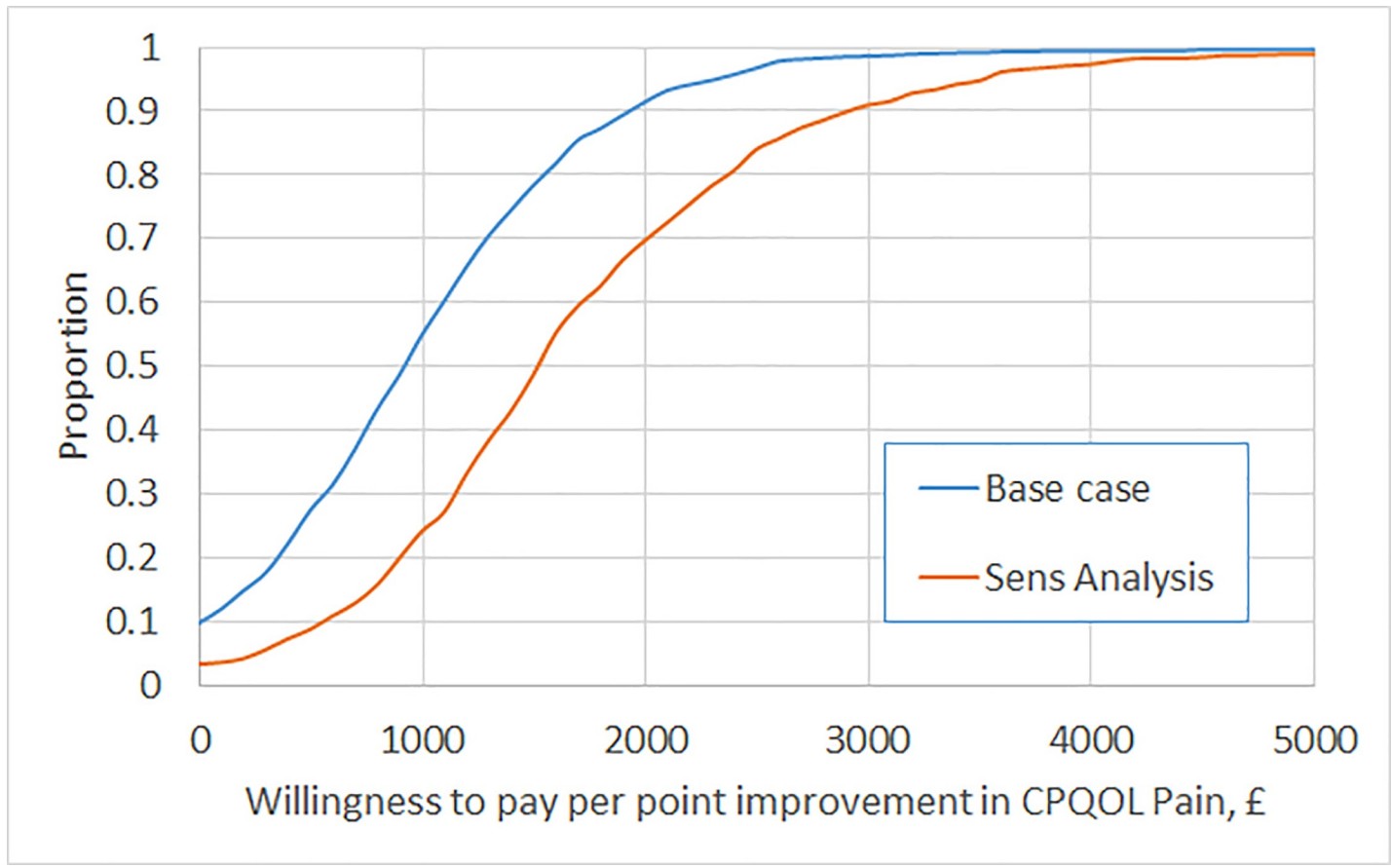

**Fig 3. Cost-Effectiveness Acceptability Curve (CEAC) for secondary outcome (CPQOL-pain).**

period of five years following surgery, and may be completely offset at ten years. Whilst the sample size from RJAH was small, uncertainty in cost-effectiveness due to sampling is low if decision makers value a unit gain in GMFM-66 at £3,000 or more. However, the use of a non-randomized comparison in the evaluation of both costs and outcomes increases the potential for bias, and introduces uncertainty into the findings that is not captured in the CEACs we report.

The judgment of whether SDR is cost-effective and the impact of sampling uncertainty in these findings depends on the value placed on the primary and secondary outcomes. In judging whether a health system should pay £1,000 to improve GMFM-66 by one point or reduce CPQOL-pain by one point, two considerations are pertinent. Firstly, both scales run from 0 to 100; one point is a very modest change. Second, our data and the available literature suggests that improvements in pain and function are maintained over time. [24] Hence the benefits of any change potentially accrue over each child's lifetime.

This is the first study to examine the cost-effectiveness of SDR. Multiple studies have reported the cost-effectiveness of intrathecal baclofen and Botox injections. The evidence indicates intrathecal baclofen is cost-effective but finds little evidence of improved outcomes with Botox. [25–30]

One previous study compared the costs of patients undergoing SDR with a matched group receiving an intrathecal baclofen pump and concluded that SDR was cheaper. [31] However, it should be noted that intrathecal baclofen is typically offered to children with greater disability

levels, functioning at GMFCS levels IV and V. Our evidence on outcomes from the CtE scheme which were reported in the clinical paper, [12] and formed the basis of the cost-effectiveness analysis is consistent with the available evidence. [32–34] A review of long term outcomes following SDR found improvements in physiology and anatomy, but noted the weakness of the evidence base. [8] A matched analysis of 13 patients undergoing SDR at a median age of 5 years with 8 patients who were eligible but did not undergo SDR did not find superior outcomes in patients receiving SDR after 10 years, but did note more frequent surgical treatment during follow-up in the non-SDR group. [35]

Evidence of the impact of SDR on costs is weaker. Cross-study comparisons of incidence of orthopaedic surgery following SDR are limited by differences in practice styles. Comparison of children undergoing SDR above and below the age of 6 years suggested a greater potential for SDR to reduce the incidence of orthopaedic surgery when was performed prior to adolescence. [36]

Our analysis has a number of strengths. Data on outcomes are taken from a large prospectively recruited sample of children which applied standardised inclusion criteria reflecting clinical evidence regarding which children could derive the greatest benefit from SDR. Missing data were limited due to CtE funding requirements to submit data and extensive efforts by the evaluators to support collection. Data on costs were taken from a centre with longstanding expertise in SDR which applied the same or tighter inclusion criteria to those of the SDR CtE scheme. Cost data included children undergoing SDR and similar children receiving supportive care from the same centre eliminating confounding from differences in practice styles or unit costs. Cost data were mostly complete to five years after assessment and data were available up to ten years after assessment.

Our analysis is limited by the lack of concurrent comparator data on outcomes and the small number of children with data on costs. Clinicians and parents of children with CP believed strongly in the benefits of SDR, rendering a randomised controlled trial unfeasible. Consequently, we undertook comparisons with historical data for GMFM-66 and a before and after comparison for CPQOL-pain collected as part of the CtE scheme. It is possible that expectations of the success of SDR biased both comparisons. Pain may have improved with conventional treatments such as Botulinum toxin. The CanChild cohort, which provided comparison data on GMFM-66 was recruited two decades ago. [18] Since that time, the management of CP has progressed and GMFM-66 trajectory in the absence of SDR may have improved. The availability of comparator data was a strength of the analysis of costs, but the RJAH cohort was small and follow-up limited beyond five years. The available data beyond five years suggests the trend of higher costs in the absence of SDR continues, indicating that our analysis may have overestimated the incremental cost of SDR. The comparator data included children declined SDR for clinical criteria. We had no *a priori* reason to believe these criteria would influence the cost of caring for these children but it is a potential confounder. Our cost data included only secondary health care. Any impact of SDR on primary health care, social care and productivity costs has not been captured. Whilst we were able to examine costs up to ten years after assessment for SDR our analysis of outcomes is restricted to a two-year follow-up period since this was the duration of CtE data collection. Finally, our analysis of cost-effectiveness ignores any potential correlation between cost and outcome data.

## Conclusion

Evidence from England suggests SDR is cost-effective. The impact of SDR on costs may be offset by reductions in the cost of supportive care over the decade following surgery. Uncertainty in this finding is lower if commissioners value a percent improvement in GMFM-66 or a

percent reduction in pain at £3,000 or more, but persists in the absence of a randomised comparison. Evidence from the SDR CtE scheme and independent cost data supports the recent decision by NHS England to commission SDR. Further research on the long-term costs and outcomes of SDR is needed to address some of the limitations in the current analysis.

## Supporting information

**S1 File.**
(DOCX)

## Author Contributions

**Conceptualization:** Robert Freeman, John Goodden, Helen Powell, Christopher Verity, Janet L. Peacock.

**Data curation:** Jennifer Summers, Bola Coker, Saskia Eddy, Karen Edwards.

**Formal analysis:** Mark Pennington, Jennifer Summers, Bola Coker, Saskia Eddy.

**Funding acquisition:** Christopher Verity, Janet L. Peacock.

**Investigation:** Muralikrishnan R. Kartha.

**Supervision:** Christopher Verity, Janet L. Peacock.

**Writing – original draft:** Mark Pennington.

**Writing – review & editing:** Jennifer Summers, Bola Coker, Saskia Eddy, Muralikrishnan R. Kartha, Karen Edwards, Robert Freeman, John Goodden, Helen Powell, Christopher Verity, Janet L. Peacock.

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
