## [Decision Letter · Decision Letter 0]

10 Mar 2020

PONE-D-19-22434

Selective dorsal rhizotomy; evidence on cost-effectiveness from England

PLOS ONE

Dear author,

Thank you for submitting your manuscript to PLOS ONE. After careful consideration, we feel that it has merit but does not fully meet PLOS ONE’s publication criteria as it currently stands. Therefore, we invite you to submit a revised version of the manuscript that addresses the points raised during the review process.

The topic of the manuscript is relevant; however, reviewers have expressed important methodological bias which make the manuscript unacceptable in its current form. I sugest the authors a deep reformulation of the manuscript according to the reviewers' suggestions in methods and consequently, in the discussion and conclusions. A new control cohort for comparison would be advisable and efforts to address analyses bias are encouraged.

We would appreciate receiving your revised manuscript by May 9th. To enhance the reproducibility of your results, we recommend that if applicable you deposit your laboratory protocols in protocols.io, where a protocol can be assigned its own identifier (DOI) such that it can be cited independently in the future. For instructions see: http://journals.plos.org/plosone/s/submission-guidelines#loc-laboratory-protocols

We look forward to receiving your revised manuscript.

Kind regards,

Inmaculada Riquelme

Academic Editor

PLOS ONE

Journal Requirements:

2) Thank you for stating the following in the Competing Interests section:

[CV has received payments or expenses for acting as the clinical chair for NHS England

Commissioning through Evaluation for SDR. MP has received personal fees from

Merck. The other authors declare no competing interests.].

3) Please include captions for your Supporting Information files at the end of your manuscript, and update any in-text citations to match accordingly. Please see our Supporting Information guidelines for more information: http://journals.plos.org/plosone/s/supporting-information.

4)  We note that you have indicated that data from this study are available upon request. PLOS only allows data to be available upon request if there are legal or ethical restrictions on sharing data publicly. For information on unacceptable data access restrictions, please see http://journals.plos.org/plosone/s/data-availability#loc-unacceptable-data-access-restrictions.

Reviewers' comments:

Reviewer's Responses to Questions

**Comments to the Author**

1. Is the manuscript technically sound, and do the data support the conclusions?

Reviewer #1: Yes

Reviewer #2: No

Reviewer #3: Yes

2. Has the statistical analysis been performed appropriately and rigorously? 

Reviewer #1: I Don't Know

Reviewer #2: I Don't Know

Reviewer #3: I Don't Know

3. Have the authors made all data underlying the findings in their manuscript fully available?

Reviewer #1: Yes

Reviewer #2: Yes

Reviewer #3: Yes

4. Is the manuscript presented in an intelligible fashion and written in standard English?

Reviewer #1: Yes

Reviewer #2: Yes

Reviewer #3: Yes

5. Review Comments to the Author

Reviewer #1: This report by Pennington et al. is a cost-effectiveness analysis of children undergoing selective dorsal rhizotomy. This current report is a subanalysis of economic and effectiveness outcomes as part of a larger clinical trial through the Commissioning through Evaluation program. The comparator group was a separate cohort of children from a previous (decades earlier) trial of Canadian children.

Overall the manuscript is quite well prepared, with clear descriptors of the methods, statistics, and analysis. Although not a strict criteria for publication, the topic of this report is quite important and impactful, as cost-effectiveness studies continue to gain importance in health system analyses. However, I do have some concerns that may benefit from consideration by the authors for clarification. Specifically:

1) I would defer to review by a health economics expert, but I question some of the terminology and methods used for this analysis. As I understand this report, the test group of children is compared to a cohort of children using previously published data (data not collected in the context of the CtE trial). In this scenario, calculation of ICERs is not appropriate, rather one should calculate the marginal cost-utility ratio. It is unclear to me whether the authors so such, including measuring the effect of treatment in each cohort in terms of a standardized metric (such as QALYs, DALYs, etc. rather than just the GMFM-66 scale or GMFCS levels) to allow for appropriate comparison across treatments.

2) The large difference in sample sizes between the cohorts really is a challenge, particularly for comparison of costs. I realize the authors addressed this in part by bootstrapping and assessment of confidence intervals, and not surprisingly the confidence intervals were quite wide. I am not sure how to best address this, and the authors do acknowledge this challenge in the description of limitations in the paper. However, I would certainly soften the conclusions of the report to recognize this limitation, as the implication is that the cost-effectiveness of SDR is clear and convincing. I am not sure this is fair to policymakers, who will interpret these statements without clear understanding of these limitations.

3) And finally, the use of such a remote historical control population (from decades earlier) is also a major limitations, both in terms of clinical effectiveness and cost analyses. The authors again acknowledge this limitation, although they should soften the conclusions accordingly. I would encourage that this data be considered pilot in nature, raising the question of whether a clinical trial (even non-randomized) would more conclusively address the question of cost-effectiveness of SDR.

Reviewer #2: This paper attempts to estimate the cost benefits of SDR. This was done by examining post-SDR outcomes (GMFM-66 and CPQOL pain score) compared to published data in the absence of surgery. The cost differential was then examined by looking at costs in an SDR group from RJAH and a group who did not have SDR in RJAH over a 10 year period from baseline assessment.

It's an interesting idea but I think that while some limitations are acknowledged, there are too many limitations and missing details to support the conclusions made.

The major flaw in this study appears to be in estimating the cost differential in those treated with SDR compared to those who were not. There area number of issues with this mainly the very small number in the non-SDR group (n=2) for whom data was available at 10 years. Looking at Figure 1 a spike in costs in the SDR group in clear at year one corresponding to the surgery. A similar spike is in the non-SDR group but no attempt is made to explain this? Was this orthopaedic surgery? Overall, there is no analysis or reporting of the standard treatment received by the control group (surgical or otherwise). Figure 1 also shows that costs are lower in the SDR group from year 3 onwards but as seen in Table 1 this 'No SDR' group were older and more involved (higher % GMFCS III and more likely to use wheelchair/buggy) so is the slightly higher costs in this very small number of children just related to that?

The benefits of SDR are established compared to published GMFM-66 prediction curves. However, this possibly over-estimates the benefits of SDR as really the benefits (or otherwise) of SDR should be when compared to standard/orthopaedic surgery which over a ten year period from baseline (at age 6-7 years) most CP children would probably have had if they did not have SDR?

There is no control data for the CPQOL pain score so it might be that the pain score improved less that it would have in the absence of surgery?

The Discussion suggests that the available literature suggests that improvements in pain and function are maintained over time. However, Munger's 2017 paper suggest no real long-term benefit in any outcome measure when comparing SDR to standard intervention and this is not mention at all.

In terms out outcome measures used here, only the pain score from the CPQOL assessment was used. Why is this and why not use other domains? This is not explained or justified.

Table 2 lists changes in GMFM 66 and CPQOL-pain over 24 months in the CtE SDR group. However, CPQOL function is also listed but not referenced at all in the paper. If listing here why not use in results? Likewise, a gait score is listed at Baseline and at 24 months but never mentioned in the paper.

Reviewer #3: The authors report on the cost-effectiveness of SDR in England.

Abstracts:

The abstract lacks the description of the underlying data set:

Age of children, number of children

Introduction: clear

Methods: the “”control”group is somewhat small, and the description of the group in table 1 is in my opinion inappropriate. Therefore, comparison of costs seems troublesome

Results

Table 2, can the authors add the number of evaluated subjects, or were data complete??

Dicussion:

The authors use a model to predict the costs of SDR as shown in figure 1. However, the follow-up was short and this assumption should get more attention in the discussion than is currently provided by the authors

“Confidence intervals are wide, and all include zero.”

(page 13)

6. PLOS authors have the option to publish the peer review history of their article (what does this mean?). If published, this will include your full peer review and any attached files.

Reviewer #1: Yes: Henry Rice

Reviewer #2: No

Reviewer #3: Yes: R. Jeroen Vermeulen, MD, PhD

---

## [Author Response · Author response to Decision Letter 0]

26 Jun 2020

Reviewer #1: This report by Pennington et al. is a cost-effectiveness analysis of children undergoing selective dorsal rhizotomy. This current report is a sub-analysis of economic and effectiveness outcomes as part of a larger clinical trial through the Commissioning through Evaluation program. The comparator group was a separate cohort of children from a previous (decades earlier) trial of Canadian children.

Overall the manuscript is quite well prepared, with clear descriptors of the methods, statistics, and analysis. Although not a strict criteria for publication, the topic of this report is quite important and impactful, as cost-effectiveness studies continue to gain importance in health system analyses. However, I do have some concerns that may benefit from consideration by the authors for clarification. Specifically:

1) I would defer to review by a health economics expert, but I question some of the terminology and methods used for this analysis. As I understand this report, the test group of children is compared to a cohort of children using previously published data (data not collected in the context of the CtE trial). In this scenario, calculation of ICERs is not appropriate, rather one should calculate the marginal cost-utility ratio. It is unclear to me whether the authors so such, including measuring the effect of treatment in each cohort in terms of a standardized metric (such as QALYs, DALYs, etc. rather than just the GMFM-66 scale or GMFCS levels) to allow for appropriate comparison across treatments.

Our analysis compared gross motor function in children receiving SDR with published data from a cohort of children who did not receive SDR. The measure of gross motor function, the GMFM-66, is the most commonly used measure. Data on children receiving SDR were taken from a National study undertaken to inform clinical decision making on the provision of SDR by the NHS in England. As was fully acknowledged in our published clinical paper, this study was not designed to include the collection of data on a comparison group of children not in receipt of SDR. Indeed, such a study is very likely to have met with ethical and practical barriers to recruitment and follow-up since the clinical and public view was firmly set at a belief that SDR is effective and that SDR may have age-sensitive outcomes. For similar reasons, it is highly unlikely that a clinical trial in which children are randomised to SDR or “standard” care without SDR will be possible due to the invasiveness of the procedure, the potential for age-sensitive outcomes and the evidence that SDR reduces spasticity. As a consequence, we compared data on changes in gross motor function over time for children in the NHS England Commissioning through Evaluation (CtE) study with published data for children with cerebral palsy who had not undergone SDR. The published data dates from before 2000 but provides information on the change in GMFM-66 as children mature according to their level of disability as assessed by Gross Motor Function Classification System (GMFCS). Growth models of the change in GMFM-66 with age according to GMFCS have been published. It is the most definitive data published on the motor function in children with CP. These data informed the assessment of the impact of SDR on gross motor function in the CtE evaluation and the subsequent report published in 2019 in the Lancet Child and Adolescent Health. 

We used the published growth models to estimate the change in GMFM-66 that we would expect to see in children who had not undergone SDR in order to derive an estimate the incremental gain associated with SDR over two years. We accept that the use of these models is inferior to a control group in a randomised trial and increases the risk of bias but we took every measure possible to minimise the bias by covariate adjustment in statistical modelling with sensitivity analyses, plus we have reported confidence intervals as for all key estimates in order to be transparent about precision. We highlight these issues in the discussion. Whilst the use of a historical control is inferior to comparison with a randomised control arm, it does not change the appropriate methodology for undertaking the economic evaluation. The incremental cost-effectiveness ratio (ICER) is in essence a measure of the marginal cost-effectiveness of an intervention. Where outcomes are quantified in QALYs or DALYs, that methodology would allow a quantification of the marginal cost-utility. We chose to undertake a cost-effectiveness analysis based on GMFM-66 primarily because we lacked data on quality of life for children not undergoing SDR, although we note that measures of quality of life and associated health state utility values required to calculate QALYs are poorly developed for children. We accept that a cost-utility analysis would have been preferable. However, cost-effectiveness analyses are frequently reported and used to aid decision making in the absence of a cost-utility analysis, and we believe that our analysis makes best use of the available data to inform decision making.

2) The large difference in sample sizes between the cohorts really is a challenge, particularly for comparison of costs. I realize the authors addressed this in part by bootstrapping and assessment of confidence intervals, and not surprisingly the confidence intervals were quite wide. I am not sure how to best address this, and the authors do acknowledge this challenge in the description of limitations in the paper. However, I would certainly soften the conclusions of the report to recognize this limitation, as the implication is that the cost-effectiveness of SDR is clear and convincing. I am not sure this is fair to policymakers, who will interpret these statements without clear understanding of these limitations.

We accept that the sample of patients available to estimate the long term impact of SDR on costs is small. We used the most appropriate techniques to accommodate the missing data and quantify the impact of the size of the sample on the uncertainty in the estimate of the impact of SDR on costs. Whilst the sample used to estimate costs had some strengths, notably the long follow-up, and comparison of data from the same centre for intervention and control, it is fair to say that the non-randomized sample introduces risk of bias. We have amended our discussion to emphasise this risk and to temper the strength of our inference. We now state:

Our analysis indicates SDR in eligible children is likely to be cost-effective if decision makers value a unit gain in GMFM-66 at more than £1400 and a concomitant improvement in CPQoL-Pain at more than £900. Children in the CtE cohort showed a greater improvement in mean GMFM-66 at two-year follow-up than values predicted using data from the CanChild cohort and bootstrap resampling indicated the finding was unlikely to be due to chance.(18) Reported pain was also significantly reduced at two-year follow-up. Data from RJAH suggests that the cost of SDR is partially offset by a reduction in the costs of caring for children over a period of five years following surgery, and may be completely offset at ten years. Whilst the sample size from RJAH was small, uncertainty in cost-effectiveness due to sampling is low if decision makers value a unit gain in GMFM-66 at £3,000 or more. However, the use of a non-randomized comparison in the evaluation of both costs and outcomes increases the potential for bias, and introduces uncertainty into the findings that is not captured in the cost-effectiveness acceptability curves we report.

We have also amended the conclusion to acknowledge the remaining uncertainty:

Evidence from England suggests SDR is cost-effective. The impact of SDR on costs may be offset by reductions in the cost of supportive care over the decade following surgery. Uncertainty in this finding is lower if commissioners value a percent improvement in GMFM-66 or a percent reduction in pain at £3,000 or more, and in the absence of a randomised comparison is the best evidence that we have. Evidence from the SDR CtE scheme and independent cost data supports the recent decision by NHS England to commission SDR. Further research on the long-term costs and outcomes of SDR is needed to address some of the limitations in the current analysis.

We have also chosen to emphasise our comparison of costs at five years after intervention. Data on resource use is rarely reported beyond five years in economic evaluations alongside clinical trials. We remain of the view that data beyond five years is a strength of our data source. However, we accept the considerable extent of missing data beyond five years for children not in receipt of SDR. For this reason we now emphasise the comparison of costs at five years and report the comparison of costs at ten years in a sensitivity analysis.

3) And finally, the use of such a remote historical control population (from decades earlier) is also a major limitations, both in terms of clinical effectiveness and cost analyses. The authors again acknowledge this limitation, although they should soften the conclusions accordingly. I would encourage that this data be considered pilot in nature, raising the question of whether a clinical trial (even non-randomized) would more conclusively address the question of cost-effectiveness of SDR.

We agree that a trial would be preferable to the CtE study to support decision making. However, as we have discussed above, there is a lack of equipoise on the clinical benefits of SDR in clinicians and parents of children with CP in the UK and worldwide and so neither of these parties would agree to participate in a randomised trial. As such, it is hard to see how such a trial will ever be undertaken. Indeed, the decision of NHS England to recommend the use of SDR following the CtE study would challenge any future attempt to collect data on children eligible for SDR but not receiving it. We accept that we cannot conclusively answer the question on the cost-effectiveness of SDR and we have highlighted this in the conclusion as described above.

However, our analysis provides the most thorough examination of the cost-effectiveness of SDR to date. It is of value to inform decision making. It is also of value in informing any future trial which might incorporate an economic evaluation.

Reviewer #2: This paper attempts to estimate the cost benefits of SDR. This was done by examining post-SDR outcomes (GMFM-66 and CPQOL pain score) compared to published data in the absence of surgery. The cost differential was then examined by looking at costs in an SDR group from RJAH and a group who did not have SDR in RJAH over a 10 year period from baseline assessment.

It's an interesting idea but I think that while some limitations are acknowledged, there are too many limitations and missing details to support the conclusions made.

The major flaw in this study appears to be in estimating the cost differential in those treated with SDR compared to those who were not. There are a number of issues with this mainly the very small number in the non-SDR group (n=2) for whom data was available at 10 years. Looking at Figure 1 a spike in costs in the SDR group in clear at year one corresponding to the surgery. A similar spike is in the non-SDR group but no attempt is made to explain this? Was this orthopaedic surgery? Overall, there is no analysis or reporting of the standard treatment received by the control group (surgical or otherwise). Figure 1 also shows that costs are lower in the SDR group from year 3 onwards but as seen in Table 1 this 'No SDR' group were older and more involved (higher % GMFCS III and more likely to use wheelchair/buggy) so is the slightly higher costs in this very small number of children just related to that?

We acknowledge the limitation in the availability of data on resource use in the control group. This was particularly acute at ten years in the control group. For this reason, we have now? chosen to present costs comparisons at 5 years as the base case in our revised manuscript. We believe the data beyond five years provides some modest further evidence of lower resource use in children who have undergone SDR compared to children who have not. However, we accept the extent of missing data limits the value of this additional information. We note that a five year follow-up is longer than that observed in many randomised trials.

We now provide clarification of treatments received by both intervention and control group children in the supplementary material in Table 1S.

We acknowledge that there were some differences in the characteristics of patients in the treatment and control groups for the cohort of children from RJAH. We controlled for differences in age and GMFCS category in regression analysis of the data. We accept that this cannot exclude the possibility that such differences may have biased the analysis but we have attempted to mitigate the impact.

The benefits of SDR are established compared to published GMFM-66 prediction curves. However, this possibly over-estimates the benefits of SDR as really the benefits (or otherwise) of SDR should be when compared to standard/orthopaedic surgery which over a ten year period from baseline (at age 6-7 years) most CP children would probably have had if they did not have SDR?

The prediction curves are based on recorded GMFM-66 scores for a cohort of children who were followed for up to ten years. These children were eligible for standard/orthopaedic surgery and would have received routine care as appropriate. They did not receive SDR. We accept that the use of this historic cohort brings some limitations and have discussed this in our revised paper. It is possible that routine care has improved since these data were collected and we acknowledge this point in the limitations section of the discussion,

‘The CanChild cohort, which provided comparison data on GMFM-66 was recruited two decades ago.(18) Since that time, the management of CP has progressed and GMFM-66 trajectory in the absence of SDR may have improved.’

There is no control data for the CPQOL pain score so it might be that the pain score improved less that it would have in the absence of surgery?

This was a major limitation of our secondary outcome measure. It is possible that the use of surgery or the use of Baclofen or Botox might have led to a reduction in pain in the control group. We have added to the limitations section of the discussion to highlight this issue. We now say,

‘Pain may have improved with conventional treatments such as Botulinum toxin.’

We had no a priori reason to believe that pain scores would improve in this population over time in the absence of surgery.

The Discussion suggests that the available literature suggests that improvements in pain and function are maintained over time. However, Munger's 2017 paper suggest no real long-term benefit in any outcome measure when comparing SDR to standard intervention and this is not mention at all.

We focussed our discussion of the literature to those papers which reported on economic evaluations of interventions for CP. The broader literature on outcomes is only briefly mentioned. The study by Munger does suggest that the benefits of SDR are limited. However, this is a small study and there are many others, some of which have more positive findings. We feel that the literature is appropriately summarised by the review we cite. Nevertheless, we now highlight this study with the following addition.

‘A matched analysis of 13 patients undergoing SDR at a median age of 5 years with 8 patients who were eligible but did not undergo SDR did not find superior outcomes in patients receiving SDR after 10 years, but did note more frequent surgical treatment during follow-up in the non-SDR group.(36)’ 

In terms out outcome measures used here, only the pain score from the CPQOL assessment was used. Why is this and why not use other domains? This is not explained or justified.

The pain score was selected as our secondary outcome measure prior to undertaking analysis of the data. We chose this particular domain because clinicians considered pain relief to be one of the primary goals of surgery, and such an impact would not be captured in our primary outcome measuring gross motor function. We now clarify this in the methods with the following addition:

‘These outcomes were selected prior to data analysis to span the main perceived benefits of SDR.’ 

Table 2 lists changes in GMFM 66 and CPQOL-pain over 24 months in the CtE SDR group. However, CPQOL function is also listed but not referenced at all in the paper. If listing here why not use in results? Likewise, a gait score is listed at Baseline and at 24 months but never mentioned in the paper.

We included this data to contextualise the changes we observed in our primary and secondary outcome measures and because we used data from those measures in the imputation models that were applied to accommodate missing data. We do not discuss these measures in the main paper because they are ancillary to the pre-planned analysis To aid clarity and focus on the cost-effectiveness analysis we have now removed this data from the table.

Reviewer #3: The authors report on the cost-effectiveness of SDR in England.

Abstracts:

The abstract lacks the description of the underlying data set:

Age of children, number of children

We now include this information for children in the national evaluation and children in the RJAH cohort providing data on costs.

Introduction: clear

Methods: the ”control” group is somewhat small, and the description of the group in table 1 is in my opinion inappropriate. Therefore, comparison of costs seems troublesome

We accept the control group was small and we highlight this point in the limitations. However, the data had strengths in that patients were observed for up to ten years and the data comes from a single centre reducing the risk of confounding from differences in clinical practice or unit costs. We used bootstrapping to quantify the impact of sampling uncertainty. The small sample size increases uncertainty in our estimates of incremental cost. We capture and report the uncertainty in both costs and outcomes in the cost-effectiveness acceptability curves. These do indeed indicate uncertainty in the likelihood that SDR is cost-effective. However, we believe they provide the best possible evidence for commissioners of services at this time.. 

Results

Table 2, can the authors add the number of evaluated subjects, or were data complete??

The table reports data for all 137 patients in the national evaluation. We report the percentage of missing observations in brackets beside each data point. Missing data was below 10% at all observation points for both the primary and secondary outcomes.

Discussion:

The authors use a model to predict the costs of SDR as shown in figure 1. However, the follow-up was short and this assumption should get more attention in the discussion than is currently provided by the authors

We were fortunate to have cost data up to 10 years on patients in the RJAH cohort. However, data were missing for a number of patients beyond 5 years and we have now amended the manuscript to report the main cost comparison on data to 5 years. This length of follow-up is longer than many clinical trials. Nevertheless, our data suggests that lower costs observed in the SDR group in the years after the initial SDR surgery compared to the comparator group persists for ten years. Hence we may have overestimated the incremental cost of SDR. We have amended the discussion to make this point explicit. We now write,

‘The availability of comparator data was a strength for the analysis of costs, but the RJAH cohort was small and follow-up limited beyond five years. The available data beyond five years suggests the trend of higher costs in the absence of SDR continues, indicating that our analysis may have overestimated the incremental cost of SDR.’

“Confidence intervals are wide, and all include zero.”

(page 13)

The purpose of our analysis was to estimate the cost-effectiveness of SDR which required estimation of the additional cost of treating children with SDR and the additional benefits of treatment. We did not set out to test whether SDR is more expensive than management of children without SDR. The confidence intervals indicate a possibility that SDR is no more expensive than management without SDR. We convey this likelihood in the cost-effectiveness acceptability curve along with the probability that SDR is cost-effective according to the value the decision maker places on improving clinical outcomes.

---

## [Decision Letter · Decision Letter 1]

15 Jul 2020

Selective dorsal rhizotomy; evidence on cost-effectiveness from England

PONE-D-19-22434R1

Dear Dr. Pennington,

We’re pleased to inform you that your manuscript has been judged scientifically suitable for publication and will be formally accepted for publication once it meets all outstanding technical requirements.

Kind regards,

Inmaculada Riquelme

Academic Editor

PLOS ONE

Additional Editor Comments (optional):

Reviewers' comments:

Reviewer's Responses to Questions

**Comments to the Author**

1. If the authors have adequately addressed your comments raised in a previous round of review and you feel that this manuscript is now acceptable for publication, you may indicate that here to bypass the “Comments to the Author” section, enter your conflict of interest statement in the “Confidential to Editor” section, and submit your "Accept" recommendation.

Reviewer #1: All comments have been addressed

Reviewer #3: All comments have been addressed

2. Is the manuscript technically sound, and do the data support the conclusions?

Reviewer #1: Yes

Reviewer #3: Yes

3. Has the statistical analysis been performed appropriately and rigorously? 

Reviewer #1: Yes

Reviewer #3: Yes

4. Have the authors made all data underlying the findings in their manuscript fully available?

Reviewer #1: Yes

Reviewer #3: Yes

5. Is the manuscript presented in an intelligible fashion and written in standard English?

Reviewer #1: Yes

Reviewer #3: Yes

6. Review Comments to the Author

Reviewer #1: (No Response)

Reviewer #3: the authors did a nice job to investigate the cost-effectiveness of selective dorsal rhizotomy for England! I think adding England does add value because health systems differ much. I agre with the previous reviewers that the control group is extremely small which is apotential bias in their calculations.

7. PLOS authors have the option to publish the peer review history of their article (what does this mean?). If published, this will include your full peer review and any attached files.

Reviewer #1: **Yes: **Henry Rice

Reviewer #3: **Yes: **R. Jeroen Vermeulen